# Oncology Drug Repurposing for Sepsis Treatment

**DOI:** 10.3390/biomedicines10040921

**Published:** 2022-04-17

**Authors:** Izabela Rumienczyk, Maria Kulecka, Małgorzata Statkiewicz, Jerzy Ostrowski, Michal Mikula

**Affiliations:** 1Department of Genetics, Maria Sklodowska-Curie National Research Institute of Oncology, 02-781 Warsaw, Poland; izabela.rumienczyk@pib-nio.pl (I.R.); mkulecka@cmkp.edu.pl (M.K.); malgorzata.statkiewicz@pib-nio.pl (M.S.); jostrow@warman.com.pl (J.O.); 2Department of Gastroenterology, Hepatology and Clinical Oncology, Centre for Postgraduate Medical Education, 01-813 Warsaw, Poland

**Keywords:** drug development, drugs repositioning, septic shock, kinase inhibitors

## Abstract

Sepsis involves life-threatening organ dysfunction caused by a dysregulated host response to infection. Despite three decades of efforts and multiple clinical trials, no treatment, except antibiotics and supportive care, has been approved for this devastating syndrome. Simultaneously, numerous preclinical studies have shown the effectiveness of oncology-indicated drugs in ameliorating sepsis. Here we focus on cataloging these efforts with both oncology-approved and under-development drugs that have been repositioned to treat bacterial-induced sepsis models. In this context, we also envision the exciting prospect for further standard and oncology drug combination testing that could ultimately improve clinical outcomes in sepsis.

## 1. Introduction

Sepsis is defined as life-threatening organ dysfunction caused by a dysregulated host response to infection [1], which can progress to a more severe form, namely septic shock [2]. In 2017, nearly 49 million sepsis cases were reported worldwide including 11 million deaths [3]. The mortality of septic shock remains high, with a recent meta-analysis for European and North American medical records estimating it as around 38% [4]. Additionally, according to the WHO epidemiological data, the burden of sepsis is likely underestimated given that up to 50% of patients with sepsis are not correctly coded using the ICD system. Therefore, the WHO in recognition of sepsis as a global health priority has issued a resolution urging the national health system to adopt a plan for strengthening efforts for the prevention, diagnosis, and treatment of sepsis [5]. The current recommendations for the treatment of sepsis are provided in the latest Surviving Sepsis Campaign guidelines [6]. However, the basic approach of sepsis treatment, which includes antibiotic therapy and supportive care, aimed at providing adequate tissue perfusion and oxygenation, has not changed for decades [7]. Despite tremendous research efforts, many clinical trials, and a better understanding of sepsis-associated pathophysiological and molecular changes, pharmacotherapies, other than antibiotics, for sepsis have not been yet implemented. The present manuscript intends to describe the preclinical efforts with oncology drug repositioning that are under development and those already used in clinics towards improving bacterial-induced sepsis. There were numerous attempts to use the oncology-indicated drugs to treat this devastating syndrome in a preclinical setup; however, a comprehensive summary of these efforts to our best knowledge has not been published.

## 2. Sepsis Overview and Failure in the Drug Development Efforts

In response to pathogen infection, the microbial-derived pathogen-associated molecular patterns (PAMPs) bind pattern recognition receptors (PRRs) on the host’s cell surface [8]. Simultaneously, infection causes the release of cellular content called damage-associated molecular patterns (DAMPs), which are also recognized by PRRs. Both PAMPs and DAMPs bind toll-like receptors (TLRs) and individual TLRs interact with different combinations of adapter proteins activating downstream signaling cascades, including mitogen-activated protein kinase (MAPK), that in turn activate various transcription factors such as interferon regulatory factors (IRFs), nuclear factor (NF)-κB, and activator protein 1 (AP-1), driving the specific immune responses [9] (Figure 1).

A profound increase in the synthesis and systemic release of cytokines initiates a series of events leading to the failure of an organism and cells to maintain physiologic homeostasis [10]. The loss of adaptive equilibrium involves, most notably, immune and neuroendocrine systems and their dysregulated interactions. These and other systemic alterations can cause a rapid loss of endothelial and epithelial barriers as well as impaired mitochondrial function, together causing organ dysfunction and often organism death [11,12].

To date, numerous small molecules and biologics have been tested in clinical trials against sepsis [13]. These therapeutics targeted known mediators of inflammation and molecular pathways altered upon the septic condition and comprises antibodies against lipopolysaccharide (LPS), compounds targeting the coagulation cascade and platelet activating factor, TLR4 and IL-1β antagonists, anti-TNFα agents, and others [13]. However, despite their preclinical efficacy in animal models of sepsis, the following clinical trials have not confirmed the beneficiary effect in septic patients [14,15]. The reasons for failures have been recently comprehensively discussed (reviewed in [15]) and herein we briefly summarize key issues that require special attention. One of the major arguments for the lack of progress in treatment translation into clinics is the use of rodents in the preclinical testing of new drugs against sepsis. Mice and humans are different in many ways, including how they respond to septic insults. For example, upon sepsis, mice usually develop bradypnea and bradycardia, while humans develop tachypnea and tachycardia. However, in retrospect, in some treatment instances, the transition to clinical testing was based on limited data from poorly designed preclinical studies utilizing rodents, which underscores that the outcome of preclinical data should be interpreted with caution [13]. Undoubtedly rodent models are an invaluable tool in deciphering molecular, cellular, and physiological events upon sepsis, which to some extent mirror human syndrome, allowing the understanding of septic pathology, and providing clues for finding treatments to improve the outcome of this devastating syndrome [16]. The enrollment of a heterogeneous groups of patients to clinical trials is another commonly recognized reason for failure. Individual heterogeneity is a significant factor in determining patient outcomes in response to pathogen infections. Patients have individual susceptibilities to infection based on comorbidities, age, and sex. While it is not possible to completely avoid individual heterogeneity, it may be possible to select patients according to the type of infection. The host response is also greatly influenced by the location of the infection. This is because different tissues contain different types of immune cells, which will respond differently to infection [15]. Despite tremendous progress in basic science and clinical studies, we have not yet developed effective new treatments for sepsis. These advances have demonstrated the complexity and heterogeneity of the syndrome, and the need to better tailor interventions to the right patient subset.

## 3. Drugs Development and Repurposing

The drug development process comprises all the activities towards transforming a compound from a drug candidate to a product approved for marketing by the appropriate regulatory authorities [17]. The first step in drug development is to identify a potential target, and the next step involves the identification of lead compounds through the drug rationale design or library compounds screening. It is followed by the medicinal chemistry-driven lead optimization to improve its pharmacological properties before the efficacy testing in preclinical models of disease. Finally, a drug undergoes clinical trials testing and eventually registration, followed by entering the market. It is estimated that de novo drug development is a 10–17 year process with an average cost of $1.8 billion and the probability of entering the market below 10% [17]. Therefore, the process is long, expensive, and risky, and many potential drugs never make it to market because of the hurdles and attrition rates along the way. As an alternative, drug repurposing or repositioning is the process of finding new therapeutic uses for drugs under a different indication. This can be done through a variety of methods, including screening drugs for new indications, studying the mechanisms of action of drugs, and using drugs in the models of diseases for which they were not originally intended. The major advantage of the drug repositioning approach is that the mechanism of action, as well as the pharmacokinetic, pharmacodynamic, and toxicity profiles of drugs, are in general well known because of the preclinical and early phase studies, therefore transition to the late phases of clinical testing can be expedited and associated cost could be substantially reduced [18]. Drug repositioning accounts for approximately 30% of the new FDA approvals [19] and there are multiple examples of successful drug repositioning (reviewed in [20]). An example of drug repurposing in the oncology field worth mentioning is Imatinib, which targets the BCR-ABL fusion protein in chronic myeloid leukemia (CML) but additionally inhibits v-kit oncogene homolog (KIT) and platelet-derived growth factor receptors (PDGFRs), which are gastrointestinal stromal tumor (GIST) oncogenes [21]. The above example of repositioning lies on the premise that drugs sometimes have off-target side effects, in addition to their established activity, that is potent enough to efficiently block altered pathways in a given disease. Accordingly, different diseases share alterations of the same molecular pathways; therefore, this raises the possibility of using the same drug for more than one disease. For example, the MAPK pathway discussed below is frequently altered both in multiple malignancies as well as in inflammatory diseases. The strategies and methods applied for drug repositioning have been comprehensively discussed elsewhere [19]. In sum, drug repurposing can be an important approach for finding new therapies for diseases that lack effective treatments, including sepsis.

## 4. Examples of Oncology Drug Repurposing for Sepsis Treatment

### 4.1. Topoisomerase 1 Inhibitors

Topoisomerase 1 (TOP1) is an essential enzyme in mammalian cells that aids in detangling the supercoiled parts of DNA that are formed during replication, recombination, transcription, and repair processes [22]. TOP1 ameliorates DNA’s topological stress by introducing a single-strand break, allowing DNA unwinding and then catalyzing the religation of the cut strand [23]. Camptothecin (CPT) was the first inhibitor of TOP1 tested in the 1970s to treat various cancers [24]. However, clinical trials with CPT were discontinued due to its low bioavailability, toxicity, unsatisfactory response rates, and, at that time, unknown mode of action [24]. A decade later, CPT’s mechanism of action has been described [25]. CPT binds to the DNA-TOP1 complex and forms a stable ternary complex that prevents religation of DNA strands and interferes with the moving replication fork, and by inducing replication arrest, it results in the lethal double-strand DNA breaks, the central mechanism for the antitumor activity of CPT [26]. The discovery of CPT as a TOP1 inhibitor has fueled the continuous development of CPT derivatives with the assumption that new derivatives will be devoid of the weaknesses of CPT and will show better anticancer properties in vivo. Topotecan (TPT) and Irinotecan (CPT-11) are CPT derivatives that were approved by the US Food and Drug Administration (FDA) as antineoplastic drugs and many CPT analogues are currently being tested in clinical trials [27] (Table 1; Figure 1).

Irinotecan is on the WHO Model List of Essential Medicines, the most important medications needed in a basic health system [41] and its indications include unresectable and metastatic colorectal cancer, platinum-resistant recurrent cervical and recurrent ovarian cancer, non-small cell lung cancer, pancreatic cancer, and glioblastoma multiforme [42,43]. TPT is used for the treatment of ovarian cancer, small cell lung cancer, and cervical cancer [27]. In their pioneering work, Rialdi et al. identified CPT as a compound that inhibited luciferase production from a reporter assay under interferon beta (IFN-β) promoter upon influenza A virus strain PR8ΔNS1 and Sendai virus infection [28]. CPT also blocked IFN-β and IFIT1 mRNA expression in A549 cells infected with PR8ΔNS1 virus, used in the concentrations negligible for A549 viability and DNA integrity. Depletion of TOP1 with siRNA in the A549 cell line followed by PR8ΔNS1 virus infection revealed significant overrepresentation of inflammatory cytokines and interferon-stimulated genes (ISGs) among downregulated genes. Chromatin immunoprecipitation (ChIP) experiments on A549 cells, revealed that TOP1 distribution on chromatin is enriched at promoters and gene bodies, and mirrors the presence of a transcriptional complex of Polymerase 2 RNA (RNAP2). TOP1 inhibition with either CPT or TPT specifically blocked RNAP2 recruitment to PAMP-inducible genes without affecting transcription of housekeeping genes. An important mechanistic observation from that study is that there is an overlap between Top1-dependent genes and those under the control of SWItch/Sucrose NonFermenter (SWI/SNF)-nucleosome remodeling complex as shown by the transcriptomic experiments with siRNA-mediated depletion of the two catalytic subunits of the SWI/SNF complex, SMARCA2 and SMARCA4 proteins. Finally, the in vivo experiments with both preventive and therapeutic interventions with CPT significantly improved survival of C57BL/6 mice during lethal endotoxic shock, *Staphylococcus aureus* infection, and the influenza virus PR8 and *S. aureus* coinfection.

### 4.2. Poly (ADP-ribose) Polymerase (PARP) Inhibitors

PARPs are essential enzymes that play a role in many cellular processes, including DNA repair, gene expression, cell death, and signaling [44]. PARPs catalyze the synthesis of the poly-(ADP-ribose) (PAR) chain from ADP-ribose moieties derived from nicotinamide adenine dinucleotide (NAD^+^), and the PAR chain can interact with a variety of proteins to regulate their function. For example, PARP activation can promote DNA repair by recruiting other proteins to the site of damage [45]. This can occur under pathophysiological stress when genotoxic amounts of reactive oxygen species (ROS) or other DNA-damaging insults cause persistent activation of PARPs. PARP activation so far has been observed in a range of pathological conditions, including diabetes mellitus, ischemia, neurological injury, vascular disease, and inflammatory diseases including sepsis [46]. The PARPs family comprises 17 members. Of them, the cellular activity of PARP1 accounts for 80–90% of NAD^+^ used by the PARP family [47]. 3-aminobenzamide (3-AB) was the first generation PARP1 inhibitor that provided evidence on the PARP cellular role related to the DNA repair mechanism; during the genotoxic stress, the cellular PARP1 abundance increases and levels of the NAD^+^ drop, triggering the mechanism responsible for DNA strands rejoining. However, the 3-AB treatment rescued NAD^+^ depletion, slowed DNA repair, and potentiated the DNA-damage-induced cell death by either DNA-alkylating agents [48] or ionizing radiation [49]. These initial observations prompted the next-generation PARP inhibitor’s development to enhance the anticancer activity of ionizing radiation and chemotherapy drugs. For example, NU1025 was a second-generation PARP inhibitor that was shown to potentiate CPT-induced DNA damage, ultimately increasing the incidence of DNA strand breaks and associated cytotoxicity [50]. The discovery that the homologous recombination-deficient tumors, for example, those driven by the tumor suppressor genes BRCA1 and BRCA2 mutation [51,52], are sensitive to the inhibition of PARP enzymatic activity, which has further fueled the development of PARP inhibitors and their ultimate implementation to clinical use. Currently, there are four inhibitors including Olaparib, Rucaparib, Talazoparib, and Niraparib that have been approved for clinical use for ovarian, breast, prostate, and pancreatic cancer and multiple oncology clinical trials with PARP inhibitors as monotherapy and in combinations are ongoing (reviewed in [44]). Although the PARP inhibitors have been developed for oncology, their efficacy has been demonstrated in many experimental models of diseases including stroke, neurodegeneration, asthma, pancreatitis, fatty liver disease, hepatitis, and sepsis (reviewed in [44]). Here we provide several examples of PARP inhibition related to sepsis and immune response as a more detailed review on PARP inhibitors use in this context has recently been published (reviewed in [53]). PARP-1 is involved in the activation of innate immune cells (macrophages, neutrophils, dendritic cells, and microglia), adaptive immune cells (lymphocytes), and the response of non-immune cells (fibroblasts, endothelial cells, and astrocytes) [54]. PARP1 acts as a transcriptional co-regulator of the NF-κB pathways and therefore regulates the production of various pro-inflammatory mediators (IL-6, pro-IL-1, ICAM-1, TNFα, COX2, inducible NOS, MIP-1 (CCL3), and MIP-2 (CXCL2)) [55]. PARP1 deficient mice upon cecal ligation and puncture (CLP)-induced sepsis exhibit significantly lower plasma levels of TNFα, IL-6, and IL-10, reduced organ inflammation, and higher survival in comparison to wild-type mice [56]. The protective role of PARP inhibitors has been shown in both LPS [29] and CLP [30] mouse models of sepsis.

### 4.3. MAPK Pathway Inhibitors

MAPKs are a family of signaling proteins that play a critical role in the regulation of many cellular processes, including inflammation [57]. MAPKs are activated in response to various stimuli, such as cytokines, growth factors, and environmental stressors, and then mediate the downstream effects of these stimuli through cytoplasmic and nuclear effectors. There are three members, which also act as the terminal kinases, of each of the major MAPK subfamilies—the extracellular signal-regulated kinase (ERK), p38, and Jun N-terminal kinase (JNK) subfamilies [58]. In a classical view, following stimulation by mitogens, cell surface tyrosine receptor kinases (TRKs), such as epidermal growth factor receptors (EGFR), activate RAS and RHO family GTPases in the vicinity of the plasma membrane through specific guanine-nucleotide exchange factors. RAS and RHO, in turn, control the activity of the kinase cascades (e.g., BRAF/MEK/ERK) that reach their targets in multiple subcellular compartments, including the nucleus [59]. Apart from phosphorylation of transcription factors and cytoskeletal proteins, the MAPKs also activate downstream protein kinases, including the p90 ribosomal S6 kinases (RSKs), mitogen- and stress-activated kinases (MSKs), MAPK-interacting kinases (MNKs), MAPK-activated protein kinase 2/3 (MK2/3), and MK5 [60]. The hyperactivation of the MAPK pathway, due to mutations and amplifications of the genes encoding components of the pathway, occurs in 40% of cancers. The most frequently mutated are *BRAF* (B-Raf proto-oncogene serine/threonine kinase), RAS family genes (*KRAS* and *NRAS*), and the TRKs (*EGFR, c-MET, c-KIT*) [58,61]. The activation of the MAPK pathway enhances the growth, survival, and metabolism of the cancer cell. Therefore, the kinase constituents of the MAPK pathway, including RAS, RAF, MEK, and ERK, became the target of intensive drug development, and some of them already are being used in clinics [58]. For example, Vemurafenib was the first BRAF inhibitor developed against V600-mutated BRAF that induced significant clinical responses in more than half of patients with previously treated BRAF V600-mutant metastatic melanoma [62].

#### 4.3.1. MEK-ERK Inhibitors

The MEK and ERK inhibitors were developed to target the aberrant MAPK-signaling pathways in RAS and BRAF mutant cancers [63]. The MEK-ERK signaling is engaged by the LPS-mediated induction of TLR4-signaling in macrophages and production of TNFα and interleukin 1β (IL-1β) during the immune responses [64,65]. The U0126 and PD98059 small molecules were the first generation of MEK inhibitors [66] that exhibited modulation of immune responses and allowed linking of the MEK-ERK axis with production of other inflammatory mediators including IL-6, IL-8, IL-12, and IL-23 in various immune cells [67,68]. The therapeutic potential of U0126 has been also shown in vivo in a murine model of LPS-induced pulmonary inflammatory responses [69]. Currently, there are four MEK inhibitors approved by the FDA, including Trametinib, Binimetinib, Selumetinib, and Cobimetinib mainly for melanoma treatment either alone or in combination with BRAF-inhibitors [70]. Of these, the therapeutic potential of Trametinib has been shown in several murine models of inflammation and infection. For example, Trametinib coadministration with LPS protected mice in a lethal endotoxin shock model [31], and in the CLP model of sepsis [32]. In the latter study, Trametinib administration reduced hypothermia, serum proinflammatory cytokines, and improved levels of liver and renal tubular injury markers. MEK inhibition resulted in ERK kinase activity reduction and a decrease in mRNA expression of TNFα, IL-1β, and IL-6 in the renal cortex of Trametinib-treated CLP mice [32]. More recently, Chen et al. showed that Trametinib attenuated edema, proinflammatory mediator production, and neutrophil infiltration in LPS-induced acute lung injury (ALI) [71]. ERK kinase is the only known substrate of MEK [72], and therefore ERK inhibition is regarded as an effective strategy of MAPK pathway deactivation and overcoming the acquired resistance to BRAF and MEK inhibitors. There are at least seven ERK inhibitors tested in clinical trials and multiple have been reported in preclinical development [73,74]. Recently, through the in vitro screen, we identified the SCH772984 compound as an effective blocker of TNFα production. SCH772984 treatment significantly improved survival in the LPS-induced lethal endotoxemia and CLP mouse models of sepsis and reduced plasma levels of Ccl2/Mcp1 [33]. Transcriptomic signatures of SCH772984 compound action across several mouse organs following the CLP challenge highlighted its influence on immune response, platelet-related signaling, extracellular matrix, and retinoic acid signaling pathways. Since the improved version of SCH772984, the MK-8353 compound, and other ERK inhibitors are being tested in oncology clinical trials [74], our study indicates that the ERK inhibitors could be considered for severe sepsis treatment.

#### 4.3.2. MNK Inhibitors

MNK1 and MNK2 are serine/threonine kinases that are activated by either ERK or p38 kinases and phosphorylate Serine 209 (Ser209) on the cap-binding eukaryotic initiation factor 4E (eIF4E), a major regulator of translation in the cytoplasm [75]. The eIF4E is regarded as an oncogene, its overexpression in in vitro and in vivo models leads to oncogenic transformation and tumor formation, and its abundance is upregulated in multiple cancers [76]. Phosphorylation of eIF4E on Ser209 promotes its tumorigenic potential and the MNKs are the only kinases known to drive this process [77]. Therefore the development of MNK inhibitors has become a promising strategy to treat tumors with aberrant activity of the MNK-eIF4E axis [78]. Currently, there are three small-molecule MNKs inhibitors, namely ETC-206 [79], eFT508 [80], and BAY1143269 [81], that have entered early phases of clinical research for the treatment of solid tumors and hematological malignancies. Apart from the role in tumorigenesis, multiple studies provided evidence on the important MNKs’ role in mediating the production of an array of pro-inflammatory cytokines, including TNFα [82,83], IL-17 [84], IL-6, MCP-1 [83], IL-8 and IL-1β [85] across immune and non-immune cells. Although the MNKs inhibitors have already been developed for two decades [78], there are few studies evaluating their potency in in vivo models of systemic inflammation including sepsis. Recently, we showed that two selective MNKs inhibitors developed by Ryvu (cpd 24 and cpd 26), significantly improved survival and suppressed symptoms in a mouse model of LPS-induced sepsis. This was accompanied by amelioration of the clinical condition of the animals and a significant reduction in levels of the TNFα and IL-6 in serum, and a decrease in eIF4E (pSer209) phosphorylation in the liver and lung [34]. In another recent study, Gao et al. showed that the MNKs inhibitor, CGP57380, administration substantially ameliorated symptoms of LPS-induced AKI [35]. CGP57380 treatment improved total cells and neutrophils and decreased the production of IL-6, TNFα, and keratinocyte-derived chemoattractant (CXCL-1) in bronchoalveolar lavage fluid. The decrease in the aforementioned inflammation biomarkers as well as the suppression of the eIF4E phosphorylation was also observed in bone marrow-derived macrophages. The engagement of MNK2 in lung injury was further confirmed in MNK2 knockout mice where LPS-challenged MNK2-deficient mice exhibited improved lung histopathological score, reduced neutrophil counts, and significantly lower IL-6, TNF-α, and CXCL-1 abundances in bronchoalveolar lavage fluid. These examples of preclinical use of MNKs inhibitors highlight the central role of these kinases in mediating signals critical for pro-inflammatory responses, therefore repositioning the MNKs inhibitors could be a novel therapeutic approach for chronic and acute inflammation, including sepsis.

### 4.4. Anaplastic Lymphoma Kinase (ALK) Inhibitors

ALK (Anaplastic lymphoma kinase) is a membrane-bound tyrosine kinase receptor, activated by ligands containing a FAM150 domain. ALK plays a crucial role in activation of multiple signaling pathways, including PI3K-Akt and MAPK pathways. ALK fusion oncogenes have been discovered in multiple cancers, including anaplastic large cell lymphoma (ALCL) and non-small-cell lung cancer (NSCLC). Point mutations within ALK kinase domain have also been identified, mainly in neuroblastomas [86]. Due to its presence in multiple cancers, ALK was considered an attractive drug target. Currently, there are five FDA approved ALK inhibitors, and while they present with significantly longer progression-free survival and response rate in NSCLC [87], they are also prone to relatively fast (less than a year) development of drug resistance [88,89]. Recently, a promising prospect of repurposing ALK kinase inhibitors as treatment for sepsis emerged. Zeng et al. [36] proposed that ALK is crucial to the pathogenesis of sepsis: it directly interacts with EGFR to trigger AKT phosphorylation and activate IRF3 and NF-κB signaling pathways, which leads to the release of cytokines and IFN-β, resulting in immune dysfunction and septic shock. After screening 464 compounds in immortalized bone-marrow-derived macrophages, Zeng et al. found that AZD3463 [90], a potent ALK inhibitor, was one of the top five compounds that blocked 3′3′-cGAMP-induced IFN-β release. Conversely, ALK was found to be among the top-ranked molecules, which promoted 3′3′-cGAMP-induced stimulator of interferon genes (STING) protein activation. Further analysis revealed that, while AZD3463 inhibits STING activation in macrophages or monocytes by triggering cell death, other ALK inhibitors, LDK378 (Ceritinib) and AP26113 (Brigatinib) do not affect cell viability. After creating stable ALK knockdown macrophage cell lines, Zeng et al. proved that ALK depletion itself does not lead to cell death. In the experimental CLP sepsis model, administration of Ceritinib resulted in survival increase and provided protective effects for most organs, affected by sepsis, including the heart, kidneys, and liver. Ceritinib improved mice survival even in high-grade sepsis. Ceritinib also reduced proinflammatory cytokine expression and serum accumulation, including IL-7, TNFα, and IFN-β. Similar protective effects were observed in LPS-induced lethal endotoxemia. Those findings were replicated, to an extent, by Ge et al. [37] on a rat CLP model. Ge et al., confirmed two-fold increase in survival rates, as well as the reduction in proinflammatory cytokine expression. They also noted an improvement in microcirculation parameters in comparison with non-treated sepsis group. These findings used only animal models, however, Zeng et al. [36] noted that similar alterations in the ALK-EGFR-AKT pathway might be present in human sepsis, as septic patients have elevated mRNA levels of ALK, EGFR, STING, and IRF3. To conclude, based on research conducted thus far, ALK inhibitors, specifically Ceritinib appear to be good candidates for repurposing for sepsis treatment. Recent research by Dayang et al. [91], performed on HUVEC cells, confirmed the role of ALK in the activation of inflammatory gene and protein expression. At the same time, ALK inhibition with Ceritinib did not have any anti-inflammatory effect in this study: NF-κB activation status and expression of pro-inflammatory cytokines were unaltered in LPS-stimulated HUVEC after treatment. Since Ceritinib [92,93] may also inhibit different tyrosine kinases under higher concentrations, further mechanistic studies are required to determine the extent of ALK’s role in lethal sepsis.

### 4.5. Immune Checkpoint Inhibitors

Immune checkpoints are mechanisms responsible for negative regulation of the immune system via regulatory T-cells (Tregs) or coinhibitory molecules. Those checkpoints are often used by developing tumors to bypass the possible autoimmune attack. Thus, overcoming this immune tolerance by blocking CTLA4 and PD1 pathways has recently become a major focus for immunotherapy in cancer [94]. Since 2011, six immune checkpoints have been approved by FDA, targeting mainly the PD1 pathway, with the most prominent being Nivolumab and Pembralizumab [95]. In murine models of experimental sepsis, both PD-1 [96] and PD-L1 [97] deficient mice were significantly protected from CLP-induced lethality, displayed reduced organ damage, a less severe cytokine storm, and in the case of PD-1 knockout, an improved bacterial clearance. Clinical studies have shown elevated levels of PD-1, which increased the risks of overall mortality and nosocomial infection [98]. Other co-inhibitory molecules were also increased during sepsis, including CTLA4 and BTLA. Their overexpression during sepsis leads to immunosuppression and causes efficient clearance of invading pathogens [99,100]. In preclinical studies, administration of either PD-1 or PD-L1 antibody improved overall survival in the CLP mouse model of sepsis [38,39,40]. It has also been demonstrated that utilization of either PD-1 or PD-L1 antibodies in donor blood restores neutrophil and monocyte function [101] and reverses T-cell exhaustion [102]. Phase I clinical trials with PD-1/PD-L1 antibodies were also conducted and improvement of patients’ biomarker values (but not overall survival) was reported [103]. The subject of targeting immunosuppression in sepsis was extensively reviewed multiple times [99,100,104] with the latest proposition focused on combining complement and immune-checkpoint inhibitors [100].

## 5. Perspective and Future Directions

Several potential new drugs for treating sepsis are currently being developed and tested in clinical trials [105]. However, the repositioning of oncology drugs to treat this syndrome has not yet become mainstream in clinical trials, except the immune checkpoint inhibitors, despite multiple encouraging results in preclinical studies with bacteria-induced sepsis (Table 1). This may change soon as the ongoing pandemic caused by severe acute respiratory syndrome coronavirus 2 (SARS-CoV-2) has expedited the drug repurposing efforts towards finding antiviral agents and relieving coronavirus induced lethal inflammation. For instance, a therapeutic treatment with TPT suppressed SARS-CoV-2-induced inflammation in hamsters [106]. Additionally, the efficacy of several MEKs inhibitors, including Trametinib, in blocking SARS-CoV–2-host interaction and production of inflammatory cytokines has been recently shown [107,108]. Furthermore, the potential in alleviating COVID-19-associated hyper inflammation has also been suggested for PARP inhibitors [109]. These examples again underscore the potential of oncology drugs for ameliorating systemic inflammation, paving the way for their broader incorporation into sepsis-centered clinical trials. However, the design of those future clinical trials should build on the lessons learned from the negative past experiences in that field, for example, the worsened outcome for patients with sepsis in the treatment arm with high doses of Etarnecept [110], or a significant survival deterioration for patients with septic shock treated with the 546C88 compound, an unselective nitric oxide synthase inhibitor [111]. This could be achieved by incorporating the effective methodology related to the selection criteria, exclusion variables, and data monitoring that would allow for rigorous detection of early benefit and early indication of harm in sepsis clinical trials [112]. Furthermore, acknowledging the heterogeneous nature of sepsis at the molecular and immune level and how the syndrome manifests itself clinically creates challenges in finding a cure that will work for everyone. We are still learning how the septic condition affects the host systematically. For example, it was surprising to uncover that despite decades of the utilization of the CLP-induced model of sepsis, the characterization of molecular changes across vital mouse organs has not been studied until recently [113]. Therefore, further basic science efforts with oncology drugs encompassing various in vitro and in vivo sepsis models are warranted to deepen the general and cell line-specific molecular mechanism of their action. The growing number of oncology drugs that have already been successfully used in the sepsis preclinical models enables the continuation of in vitro and in vivo testing of drug combinatorial dose responses that could enable future interventions that ultimately would improve clinical outcomes in sepsis. A specific treatment for sepsis that is much awaited and needed will hopefully emerge soon.

## Figures and Tables

**Figure 1 biomedicines-10-00921-f001:**
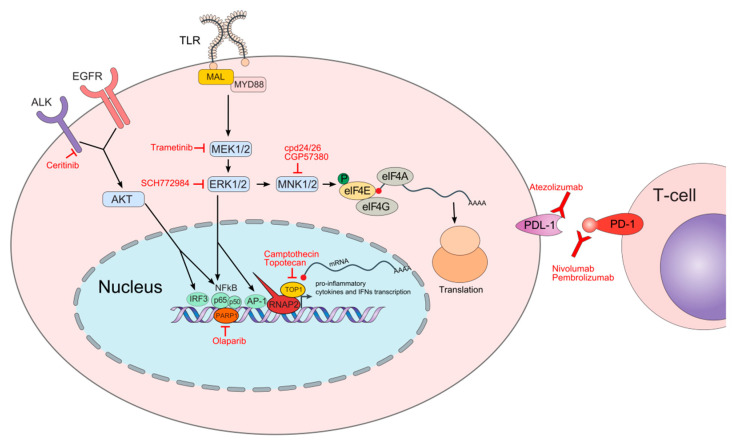
Schematic depiction of molecular pathways and processes involved in cellular responses upon pathogen infection and respective oncology drugs repurposed for experimental sepsis treatment. The names of oncology drugs discussed herein are in red font next to their molecular targets. Following toll-like receptor (TLR) activation, the myeloid differentiation primary response protein 88 (MYD88) together with MYD88 adaptor-like protein (MAL) are recruited to the TLRs initiating the cascade of molecular events activating mitogen-activated protein kinase (MAPK) components including MEK and extracellular signal-regulated kinase (ERK) that ultimately mobilize chromatin recruitment of interferon regulatory factors (IRFs), nuclear factor (NF)-κB, and activator protein 1 (AP-1) to gene loci initiating expression of the specific immune responses. ERK also indirectly influences translation by regulating MAPK-interacting kinase (MNK) that phosphorylates the eukaryotic translation initiation factor 4E (eIF4E) at Ser209 from a cap-binding complex leading to the translation of transcripts encoding pro-inflammatory cytokines, including tumor necrosis factor (TNF)α. Anaplastic lymphoma kinase (ALK) participates with the epidermal growth factor receptor (EGFR) to promote AKT stimulation, which then activates NF-kB and IRF3 factors to induce the expression of proinflammatory cytokines and interferon β (IFN β). In the nucleus, the poly (ADP-ribose) polymerase 1 (PARP1) acts as a transcriptional co-regulator of the NF-κB transcriptional factor while the topoisomerase 1 (TOP1) facilitates polymerase 2 RNA (RNAP2) recruitment to the genes encoding pro-inflammatory mediators. Checkpoint proteins including programmed cell death protein 1 (PD-1) and PD-1 ligand (PDL-1) play an essential role in transitioning from a hyper- to hypo-inflammatory response. Both PD-1 and PDL-1 are expressed on immune cells while PDL-1 is also expressed on non-immune cells.

**Table 1 biomedicines-10-00921-t001:** List of established and under-development oncology therapeutics effective in experimental models of sepsis.

Host Protein	Name of the Compound	Oncology Indication	Sepsis Model	Reference
TOPO1	Topotecan Camptothecin	ovarian cancer, small cell lung cancer, cervical cancer	LPS *S. aureus* infection	[28]
PARP	Olaparib	ovarian cancer, breast cancer, prostate cancer, and pancreatic cancer	CLP LPS	[29,30]
MEK1/2	Trametinib	melanoma	CLP LPS	[31,32]
ERK1/2	SCH772984	melanoma colon cancer	CLP LPS	[33]
MNK1/2	Cpd 24/26 CGP57380	breast cancer, colorectal cancer, diffuse large B cell lymphoma	LPS	[34,35]
ALK	Ceritinib	non-small-cell lung cancer	CLP LPS	[36,37]
PD-1 PD-L1	Anti PD-1 Anti PD-L1	cancers with high tumor mutational burden	CLP	[38,39,40]

LPS; lipopolysaccharide, CLP; cecal ligation and puncture.

## Data Availability

Not applicable.

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
