# Peer review of "Oncology Drug Repurposing for Sepsis Treatment"

_biomedicines, 2022, doi:10.3390/biomedicines10040921_

Round 1
Reviewer 1 Report
The authors review how the development of oncology drugs could help for sepsis treatment. The paper is very interesting and seems quite complete to me. The paper is well written and pleasant to read.
My only comment is that more illustrations would increase the impact of the paper: ideally a figure including the various interventions schematically; otherwise an additional Table would be welcome.
Author Response
Response: We are contented that the Referee finds our review pleasant to read. In the revised version, we include a figure that depicts the molecular pathways and processes that are targeted with oncology drugs and have shown efficacy in experimental models of sepsis.
Reviewer 2 Report
The authors propose a very extensive review on the new therapies of sepsis, with agents borrowed from cancer therapy. The review is very long and verbose, and not easy to read for a clinical reader to whom it is aimed as a target. The authors start from the premise that in the last decade of the last century all the large trials that have examined the various inhibitors of inflammatory mediators (NSAIDs, nitric oxide, Platele activating factor, various interleukins, TNF, the same lipopolysaccharide) have failed, and have not shown clinical utility. Major concerns 1) the authors include coticosteroids among the failed drugs in the treatment of sepsis. This is not correct to me as a statement. 2) The drugs proposed in this review work at a deeper level, and they enter metabolic pathways common to many stimuli. The data brought to support their usefulness in sepsis are only experimental, in animals and in well-defined sepsis models (LPS, CLP). The authors must mitigate their enthusiasm by taking into account the above considerations on similar negative past experiences, also subject to severe criticism on the ethics of trials that have shown an increase in mortality in the treated arm (to give an example, trials on inhibitors nitric oxide).3) the Authors must shorten the text, and insert one or more figures for each sub-chapter, which make clear the potential of the drug in sepsis. 4) The authors must divide Tab. 1 into different tables, appropriate for each sub-chapter examined
Author Response
- the authors include coticosteroids among the failed drugs in the treatment of sepsis. This is not correct to me as a statement.
Response: we excluded coticosteroids from the sentence about failed drugs
- The drugs proposed in this review work at a deeper level, and they enter metabolic pathways common to many stimuli. The data brought to support their usefulness in sepsis are only experimental, in animals and in well-defined sepsis models (LPS, CLP). The authors must mitigate their enthusiasm by taking into account the above considerations on similar negative past experiences, also subject to severe criticism on the ethics of trials that have shown an increase in mortality in the treated arm (to give an example, trials on inhibitors nitric oxide).
Response: We agree with this comment. To address Referee’s suggestion we further discuss in the last paragraph that any future clinical trials in sepsis with oncology drugs should take into account the previous failures and apply rigorous methodology to mitigate the harm to patients enrolled in sepsis trials.
3) the Authors must shorten the text, and insert one or more figures for each sub-chapter, which make clear the potential of the drug in sepsis.
4) The authors must divide Tab. 1 into different tables, appropriate for each sub-chapter examined
Response to 3 and 4: We agree with Referee #2 that the content in our review may be of less interest to clinicians who expect more practical advice on treating sepsis patients. However, our review is aimed at a wider audience, including those who expect new information on the mechanisms underlying sepsis. Therefore, we believe that the shortening of the manuscript would detract the reader from the molecular insights of oncology drugs action that we find important to acknowledge for their transition to clinical trials. We also believe that the inclusion of separate figures and tables for each subchapter would obscure the presentation. However, as suggested by both Referees we included a figure that depicts the molecular pathways and processes that are targeted with oncology drugs and have shown efficacy in experimental models of sepsis.
Round 2
Reviewer 2 Report
The paper is improved. However, in my opinion the paper is clearly experimental.